# Effectiveness of formal onboarding for facilitating organizational socialization: A systematic review

**Elin Frögéli** *, **Bo Jenner, Petter Gustavsson**

Department of Clinical Neuroscience, Division of Psychology, Karolinska Institutet, Solna, Sweden

* elin.frogeli@ki.se

## Abstract

### Objective

To investigate the effectiveness of formal onboarding programs and practices for new professionals.

### Introduction

New professionals may experience high levels of stress and uncertainty. Formal onboarding programs and practices aim to facilitate the socialization of new professionals by structuring early experiences. However, there is a lack of evidence-based recommendations of how to onboard new professionals.

### Methods

This review considered studies that compares the effect of formal onboarding practices and programs for new professionals between 18–30 years of age (sample mean) to the effect of informal onboarding practices or 'treatment as usual' in professional organizations internationally. The outcome of interest for the review was the extent to which new professionals were socialized. The search strategy aimed to locate both published studies (dating back to year 2006) and studies accepted for publication written in English using the electronic databases Web of Science and Scopus (last search November 9 2021). Titles and abstracts were screened and selected papers were assessed by two independent reviewers against the eligibility criteria. Critical appraisal and data extraction were performed by two independent reviewers using Joanna Briggs Institutes templates. The findings were summarized in a narrative synthesis and presented in tables. The certainty of the evidence was assessed using the grading of recommendations, assessment, development and evaluations approach.

### Results

Five studies including 1556 new professionals with a mean age of 25 years were included in the study. Most participants were new nurses. The methodological quality was assessed as low to moderate and there were high risks of bias. In three of the five included studies, a

---

**Data Availability Statement:** All relevant data are within the paper and its Supporting Information files.

**Funding:** The work was founded by Afa Insurance [grant number: 180292]. The funders had no role

in study design, data collection and analysis, decision to publish, or preparation of the manuscript.

**Competing interests:** The authors have declared that no competing interests exist.

statistically significant effect of onboarding practices and programs on new professionals' adjustment could be confirmed (Cohen's d 0.13–1.35). Structured and supported on-the-job training was shown to be the onboarding strategy with the strongest support to date. The certainty of the evidence was rated as low.

## Conclusion

The results suggests that organizations should prioritize on-the-job training as a strategy to facilitate organizational socialization. For researchers, the results suggest that attention should be given to understanding how to best implement on-the-job training to ensure strong, broad, and lasting effects. Importantly, research of higher methodological quality investigating effects of different onboarding programs and practices is needed.

**Systematic review registration number:** OSF Registries osf.io/awdx6/.

## Introduction

Organizational socialization is defined as "the process through which individuals acquire the knowledge, skills, attitudes, and behaviors required to adapt to a new role" ([1], p. 3). Socialization occurs at multiple times throughout an individual's career, whenever there is a change in the role of some sort, either one is transferring from an education into a profession, from one organization to another organization, or from one role to another within the same organization [1]. However, it has been proposed that the transition from school to work is more challenging than later transitions [2]. New professionals may experience high levels of stress and uncertainty as they are exposed to situations that are characterized by unpredictability and uncontrollability, and perceived as socially risky [3–5]. In relation to these stressors, the primary goals of new professionals are to develop a sense of competence and control, predictability and meaningfulness, and social belonging [5, 6].

Onboarding refers to "all formal and informal practices, programs, and policies enacted or engaged in by an organization or its agents to facilitate newcomer adjustment" ([7], p 268). The practices, programs, and policies are put in place by management and/or HR departments and aim to structure newcomers' early experiences, which is expected to facilitate their organizational socialization [7].

Organizations that are more active and more effective in onboarding new employees enjoy a 2.5 times as great revenue growth and a 1.9 times as great profit margin than organizations that are less active and effective in relation to onboarding their newcomers. As such, attending to the onboarding of new employees was one of six top HR practices that differentiated companies that were high performing economically from those less so according to an investigation by Strack et al. [8]. Furthermore, results of a meta-analysis by Crook et al. showed that human capital resources are significantly related to firm performance [9]. The authors suggests that, "to achieve high performance, firms need to acquire and nurture the best and brightest human capital available and keep these investments in the firm"([9], p. 453).

In describing what it is that organizations do (or ought to do) to introduce their new employees, a number of models have been proposed. The most influential models are the socialization tactics model [10], the Inform-Welcome-Guide (IWG) model [11], and the Four C's (compliance, clarification, culture, and connection) model [12]. Below, we describe the IWG model as this was developed based on a review of onboarding and thus provides an overview of the field.

The IWG model suggests that socialization practices may be organized based on their purposes that fall into one of three categories: to provide new professionals with information (through communication, resources, or training), to make newcomers feel welcomed into the new organization, or to provide the new professional with guidance. Practices within the inform category aim at helping new professionals learn what they need to know to function in their new role. Welcome-practices aim at addressing the emotional needs of new employees and facilitating the development of social relations. Finally, practices within the guide category aim at facilitating the new professionals' transition from an outsider to an insider by active and direct assistance.

In addition to these models for conceptualizing onboarding practices, a number of researchers have listed the practices that they consider to be important. Accordingly, the newcomer should be given a personalized welcome, a tour of the facilities, the workplace should be prepared for the arrival of the new employee and practical information such as abbreviations used within the organization, names, and contact information of important people in the organization should be communicated in writing [7, 11–14]. Furthermore, onboarding practices and programs should be formal and participatory, information about the objectives and time line, as well as information about the new professional role, expectations, responsibilities, opportunities for development, and the organizational culture should be clearly communicated [12, 15]. Furthermore, the newcomer should be given on-the-job training, be encouraged to monitor co-workers, be assigned a mentor, coach, or 'buddy', and be given the opportunity to meet with key stakeholders within the organization [7, 11, 13, 14]. Finally, progress should be monitored with regular check-ins during the first year [12].

The key indicators of new professionals' socialization are role clarity, task mastery, and social acceptance [2]. Role clarity concerns having enough information about what is expected of oneself as a consequence of the role one is in, and sufficient knowledge of what behaviors are appropriate to achieve the goals one is expected to achieve. When one is not clear about one's role expectations, it is difficult to direct attention and effort in a successful manner, which results in confusion and lowered motivation and performance [16]. Role clarity is sometimes measured as its flipside, role conflict [17, 18] or role ambiguity [18, 19]. Task mastery refers to the newcomers' experiences of being able to manage tasks effectively. New professionals that perceive themselves as able to effectively manage tasks are expected to put more goal directed effort into the execution of tasks and endure in the face of difficulties [20]. Other constructs that are used as measures of a sense of competence and control are self-efficacy [2, 3] learning [21], and competence [22]. Finally, social acceptance refers to the new professionals' inclusion into their new group of colleagues and their experiences of social support. The experience of being included in the new social group is expected to affect the degree to which new professionals experience situations to entail social risks [16]. Perceived social risks have been found to reduce newcomers' engagement in proactive behaviors, a class of behaviors that, if enacted, would be expected to contribute to adjustment [23]. New professionals' experiences of being included in the new social group is also referred to as work-group integration [16] and social integration [17, 24].

Studies show that onboarding practices are related to greater adjustment. Results of two meta-analyses show that treating a number of newcomers in a standardized way, clearly distinguishing them from other employees and exposing them to a special set of discrete, predefined, experiences within an explicit plan and time frame, as well as engaging experienced members of the organization to function as role models, while still encouraging the newcomers to be who they are, is associated with higher levels of the adjustment indicators [2, 18]. Specifically, introducing newcomers to an organization this way is associated with higher levels of

newcomers' role clarity and role orientation [2, 18], lower levels of role ambiguity and role conflict [18], higher levels of task mastery [2], and finally, higher levels of social acceptance [2].

In sum, there is considerable theoretical and practical knowledge available concerning onboarding of new employees [2, 11]. However, the available reviews [2, 18] were published almost 15 years ago, they did not exclusively include longitudinal data, and there is still a lack of understanding as regards the effectiveness of onboarding practices and programs [14, 25]. A preliminary search of the International prospective register of systematic reviews (PROSPERO), Open Science Framework (OSF) Registries, Web of Science, the Cochrane Database of Systematic Reviews, and Joanna Briggs Institutes (JBI) Evidence Synthesis has been conducted and no current or underway systematic reviews on the topic could be identified. Thus, the objective of this systematic review was to investigate the effectiveness of formal onboarding for facilitating new professionals' organizational socialization. We built our work on the definition of onboarding by Klein and Polin [7]. Below are the definitions use to refer to some central terms:

*Formal onboarding practices*–any organization- or researcher-initiated formal onboarding activity for facilitating the adjustment of new professionals (e.g., inviting new professionals to engage in a training session). Onboarding practices are generally expected to take place on single occasions but may also be recurring.

*Formal onboarding programs*–any organization- or researcher-initiated formal program for facilitating the adjustment of new professionals. Onboarding programs are expected to extend over weeks or months and include a number of onboarding practices.

*Informal onboarding*–all activities enacted or engaged in informally by any member of the organization to facilitate the adjustment of new professionals (e.g., a colleague clarifying to the new professional how a procedure is performed).

*Treatment as usual*–formal onboarding practices or programs that are used as comparison conditions in studies. An example study may evaluate the addition of a formal onboarding practice to an established program. The experimental group is exposed to the established program plus the additional component, and the comparison group is only exposed to the established program.

*New professionals*–professionals who have newly entered the labor market (see terms used in the search strategy below).

## Review questions

The review focused on two research questions, namely:

1. What is the effect of formal onboarding practices vs informal onboarding or treatment as usual on new professionals' socialization?

2. What is the effect of formal onboarding programs vs informal or treatment as usual onboarding on new professionals' socialization?

## Methods

The study was conducted in accordance with JBI methodology for systematic reviews of effectiveness evidence [26]. The review protocol was registered in OSF Registries (osf.io/awdx6/).

## Inclusion and exclusion criteria

The following inclusion and exclusion criteria were used:

1. Population: New professionals with a sample mean age of 18–30 years. Age was used as a proxy for time since labor market entry as we expected the actual time since entry would seldom be reported.

2. Intervention: Any formal onboarding practice or program in professional organizations internationally. Programs or practices starting later than three months into the new professionals' employment, and programs or practices that are part of professional education in practice (e.g., as part of medical residencies) were excluded.

3. Comparison: Informal onboarding practices or treatment as usual (as defined above).

4. Outcome: The extent to which new professionals are socialized. This was operationalized using the key indicators of socialization–role clarity, task mastery, and social acceptance [2]–and related constructs (e.g., role conflict, role ambiguity, workgroup integration, social integration, self-efficacy, confidence, learning, knowledge, and performance). The following scales were also considered as measures of socialization: the Content Areas of Socialization (CAS [18]), the Organizational Socialization Inventory (OSI [27]), the Employee Adjustment Survey/Socialization Knowledge Measure (EAS [28]), the Newcomer Socialization Questionnaire (NSQ [21]), and the Newcomer Understanding and Integration Scale (NUIS [29]). Additional measures identified through the search process were included. No prioritization was made between measures.

We adhered to the JBI recommendations as regards type of studies [30]. In accordance with these, we aimed to include randomized controlled trials (RCTs) and, in the absence of such trials, quasi-experimental studies and observational studies. We only considered observational studies with some form of between-groups design (i.e., cohort studies and case-control studies, thus excluding cross-sectional studies, before-and-after studies, and case series). Only studies published in English were considered.

## Search strategy

The search strategy aimed to identify papers that were either published or accepted for publication in scientific journals. As this work builds on the results of a previous meta-analysis [2], the time frame considered extended from January 2006 until November 2021.

A search string for use in electronic databases was developed in collaboration with a university librarian at Karolinska Institutet University Library with expertise in systematic literature searches. Initial lists of search terms for "new professional" and "formal onboarding" were developed by the researchers. These lists were reviewed by a group of Swedish experts in the field of onboarding and a series of additional terms were added. Finally, in the process of developing the final search strategy for the electronic databases, a few additional terms were added based on their use in key papers. The list of search terms used for the method component was developed based on the guidelines in the JBI manual for systematic reviews of effectiveness. The search was performed using the electronic databases Web of Science and Scopus. The search strategies for each database are included in Appendix I in S1 Appendix. The search was performed in April 2021 and then completed by an additional search in November 2021.

Google Scholar was used with a modified search strategy and the first 200 hits were investigated for inclusion. In addition, the trial registry OSF Registries (https://osf.io/registries) was searched using the search terms "socialization" and "onboarding".

The reference lists of any paper found eligible was examined for identification of possibly relevant papers. In addition, using the citation index in Web of Science, additional articles citing any of the papers found eligible were examined. The conference programs of the annual meetings of the Society for Industrial and Organizational Psychology dating back to 2006 was investigated and authors of relevant presentations were contacted and asked for additional papers. Finally, the first author of the papers found eligible for inclusion were contacted and asked about additional trials that may fit the criteria.

## Study screening and selection

After removal of duplicates, the selection process was carried out in three steps: 1) identification of potentially relevant papers based on titles; 2) identification of potentially relevant papers based on abstracts; 3) identification of relevant papers based on full texts, evaluated against the criteria of eligibility using a standardized table in Microsoft Office Excel. In the case of papers that were found possibly eligible but were lacking in the reporting of information, the first author was contacted and asked to complete data (maximum of three e-mail attempts). Step one was performed by EF and step two was performed by EF, BJ, and PG investigating each abstract in pairs. Any paper found suitable for step three by any one of EF, BJ, or PG was included. Step three was performed in the same way as step two. A PRISMA flow chart [31] was developed to illustrate the process. The software Rayyan [32] was used for keeping records in the evaluation of titles and abstracts.

## Critical appraisal

A quality assessment was performed on all papers found suitable for inclusion. The assessment of methodological quality was performed by EF and BJ using the standardized critical appraisal checklists for RCTs, quasi-experimental studies, and observational studies [30] in Microsoft Office Excel. Any disagreements were discussed by all three reviewers until consensus. The results of the critical appraisal were used for the critical examination of the impact of the methodological quality of the studies on the results of the review. No papers were excluded from the synthesis based on the results of the quality assessment.

## Data extraction

A standardized data extraction table was used in Microsoft Office Excel. Information about the design, population, setting, onboarding program/practice, comparison condition, and outcomes were extracted from the full-text version of papers. The extraction was performed independently by EF and BJ and any disagreements were discussed by all three reviewers until consensus. The data extracted from each paper was sent to the corresponding authors of each paper who was asked to point out any mistakes made in the data collection process, complete missing data, or confirm that the extraction was performed correctly (maximum of three e-mail attempts).

There were no cases of multiple publications for the same study. In the case of multiple time points of data collections and multiple measures of socialization, we included post-intervention data for all relevant outcome measures in the table of results but focus our narrative synthesis and discussion on the measurements that best fit the research questions. The focus on post-intervention data is chosen for the purpose of comparison.

## Data synthesis and assessment of certainty in the findings

Data synthesis was performed in line with recommendations [26]. A narrative synthesis is presented. The grading of recommendations, assessment, development and evaluations (GRADE) certainty rating [33] was used to evaluate the strength of the body of evidence. In the GRADE system, quality of evidence is rated high, moderate, low, or very low. The rating starts of assuming a set of only RCT studies of good quality which is rated as high-quality evidence. If this assumption does not hold because of study limitations, inconsistency in results between studies, indirectness of evidence, imprecision, or reporting bias, the rating of the quality of evidence is decreased. In addition, the grading may be increased if the magnitude of effects in the

included studies is very large, if there is evidence of dose-response effects, or if there is high plausibility of underestimation of effects due to omitted confounders.

## Results

The selection process is presented in the PRISMA flow chart [31] in Fig 1. A total of 14743 records were identified through the electronic databases, Google Scholar, and the OSF Registers. Of these, 5187 were duplicate records that were removed before screening. An additional 346 records were identified through investigation of the Society for Industrial and Organizational Psychology annual conference programs as well as through investigation of forward and backward citation of the papers found eligible for inclusion in the review. The number of unique records was thus 9902. Screening based on title and abstract resulted in the exclusion of 9759 records (i.e. based on a review of title and/or abstract, it was clear that the records would not fit the inclusion criteria). Of the remaining 143 records, 138 were excluded following examination of full-texts against the inclusion criteria. All studies excluded following full-text investigation are presented in Appendix II in S1 Appendix together with reasons for exclusion. The methodological quality of the remaining studies was examined and, as per protocol, no studies were excluded based on the critical appraisal. Thus, the final number of included studies was five.

## Methodological quality

The methodological quality of the included papers was investigate using the JBI-Critical appraisal checklists. Based on the questions of effects of onboarding strategies or programs in professional settings, the most important methodological aspects for the review concerned the

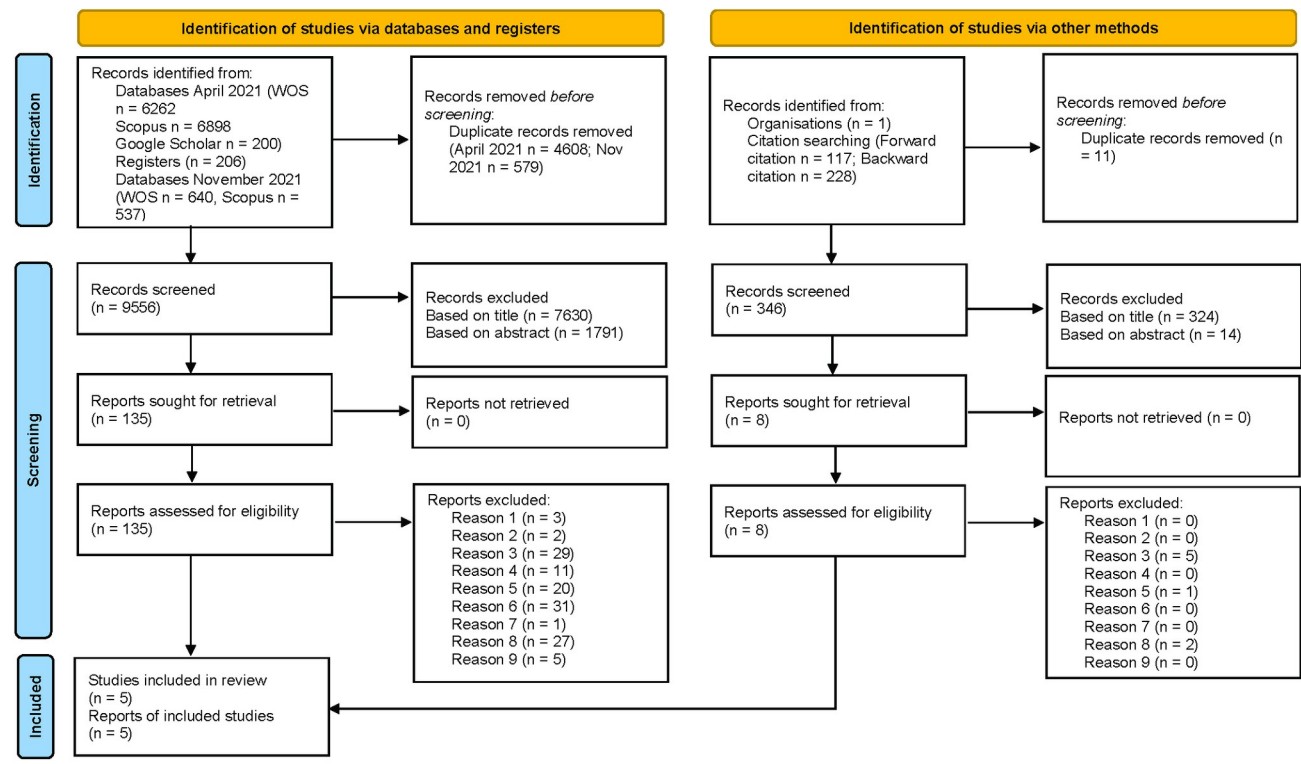

**Fig 1. Prisma flow chart.**

comparability of the intervention and the control groups, the control over other potential influences during the period of study, the reliability of outcome measures, as well as dropout from analyses. In summary, the methodological quality of the included studies was considered low to moderate. The overall risk of bias was considered high for selection bias, performance bias, detection bias, and attrition bias, and unclear for reporting bias. The results are presented in Table 1.

A potential for selection bias was present in all included studies as participants were not randomized to study conditions [34–36] or, when they were randomized, there were limitations in the methods used [37, 38]. A risk of performance bias and detection bias was present in all studies as interventionists were not blinded to treatment conditions and all studies used self-reported data for outcome evaluation. Follow-up was complete in one study [37] and in two studies missing data were imputed [34, 38] and thus there was a low risk of attrition bias. However, in the remaining studies [35, 36] data was not complete and therefore the risk of attrition bias in these studies was high. Finally, the risk of reporting bias was unclear in all studies but [38] as there were no preregistered study protocols.

## Characteristics of included studies

The characteristics of the included studies are presented in Table 2. Two of the included studies are RCTs, two are between-group designs without randomization (one of these was conducted as an RCT but in the present review data is included from a secondary analysis that do not adhere to the principles of randomization), and the fifth included study is an observational study with a longitudinal design. The total number of participants was 1556 and the mean age was 25 years. Data was collected at baseline (within the first three months following professional entry) and then again at post-intervention (7 weeks to 9 months following baseline). In [34] follow-up data was also collected at 9 and 12 months after baseline.

In four of the included studies, the study populations were new nurses [34, 35, 37, 38]. In [36], the study population included new professionals within diverse occupations such as accountancy, auditing, banking, software, chemical and manufacturing industries.

Three of the included studies [36–38] investigated effects of onboarding practices according to the definitions used in this review (simulation training, on-the-job training, stress prevention and proactivity reinforcement). The remaining two [34, 35] investigated effects of onboarding programs (transition-to-practice program, clinical training program). The formats of the practices were individual face-to-face instructor-led situated and simulated support sessions, instructor-led group didactic- and workshop sessions, and on-the-job training (self-reported). The reported doses varied from 6–12 hours [37] to nine hours [38] and the reported durations ranged from 4 weeks to three months [37, 38]. The frequency of the activities was every other week in [38] and was not reported for the other programs. The content of the practices included training in situated and simulated commonly occurring clinical situations, didactics about organizational socialization, stress, and recovery, behavior change techniques for increased proactivity, and on-the-job training.

The format of the programs were online course and preceptorship [34] as well as ward-based clinical training with support of preceptor, classroom-based lectures with discussions and group works [35]. The doses of the programs were not reported but the duration was six (plus six) to nine months [34, 35]. The frequency of the program activities was not reported. The content of the programs included orientation, didactics, communication and teamwork, skills training, feedback and reflection [34, 35].

The control conditions included self-directed learning or clinical training without standard program [35, 37], a transition-to-practice programs without a structured curriculum [34], and

**Table 1. JBI critical appraisal checklist for RCTs, quasi-RCTs and cohort studies.**

| Critical appraisal item content | Paper 1 | 2 | 3 | 4 | 5 |
|---|---|---|---|---|---|
| Was true randomization used for assignment of participants to treatment groups? | Y | N | n.a | n.a | n.a |
| Was allocation to treatment groups concealed? Concealing the allocation sequence from those assigning participants to intervention groups | U | Y | n.a | n.a | n.a |
| Were participants blind to treatment assignment? | N | Y | n.a | n.a | n.a |
| Were those delivering treatment blind to treatment assignment? | N | N | n.a | n.a | n.a |
| Were outcomes assessors blind to treatment assignment? | N | Y | n.a | n.a | n.a |
| Was there a control group? | n.a | n.a | n.a | Y | Y |
| Were the exposures measured similarly to assign people to both exposed and unexposed groups? | n.a | n.a | Y | n.a | n.a |
| Were treatment groups similar at baseline? | Y | Y | n.a | n.a | n.a |
| Were the participants included in any comparisons similar? | n.a | n.a | n.a | N | N |
| Were the two groups similar and recruited from the same population? | n.a | n.a | Y | n.a | n.a |
| Were treatment groups treated identically other than the intervention of interest? OR Were the participants included in any comparisons receiving similar treatment/care, other than the exposure or intervention of interest? | Y | Y | n.a | U | U |
| Is it clear in the study what is the 'cause' and what is the 'effect' (i.e. there is no confusion about which variable comes first)? | n.a | n.a | n.a | Y | Y |
| Was the exposure measured in a valid and reliable way? | n.a | n.a | Y | n.a | n.a |
| Were the groups/participants free of the outcome at the start of the study (or at the moment of exposure)? | n.a | n.a | U | n.a | n.a |
| Was follow up complete and if not, were differences between groups in terms of their follow up adequately described and analyzed? OR Was follow up completed and if not, were the reasons to loss to follow up described and explored? | Y | Y | N | Y | N |
| Were outcomes measured in the same way for treatment groups? OR Were outcomes of participants included in any comparisons measured in the same way? | Y | Y | n.a | Y | Y |
| Were outcomes measured in a reliable way? | U | N | Y | Y | N |
| Was the follow up time reported and sufficient to be long enough for outcomes to occur? | n.a | n.a | Y | n.a | n.a |
| Were participants analyzed in the groups to which they were randomized? | Y | Y | n.a | n.a | n.a |
| Were there multiple measurements of the outcome both pre and post the intervention/exposure? | n.a | n.a | n.a | N | N |
| Was appropriate statistical analysis used? | Y | Y | Y | Y | Y |
| Was the trial design appropriate, and any deviations from the standard RCT design (individual randomization, parallel groups) accounted for in the conduct and analysis of the trial? | Y | Y | n.a | n.a | n.a |
| Were confounding factors identified? | n.a | n.a | Y | n.a | n.a |
| Were strategies to deal with confounding factors stated? | n.a | n.a | Y | n.a | n.a |
| Were strategies to address incomplete follow up utilized? | n.a | n.a | N | n.a | n.a |

Note: Paper 1 = Chen et al., 2017; Paper 2 = Frögéli et al., 2020; Paper 3 = Kowtha, 2011; Paper 4 = Spector et al., 2015; Paper 5 = Horii et al. 2021; Y = Yes, N = No, U = Unsure; n.a = not applicable.

**Table 2. Characteristics of included studies.**

| Study | Design | Population | Setting | Onboarding program/practice | Comparison | Outcomes |
|---|---|---|---|---|---|---|
| 1 | RCT | New nurses. BSc. No prior internship. n: T 31; E 16; C 15 Age (M): 22.32 Gender F: 100% | One hospital, different work unit Taiwan Year 2014. | Situated and simulated nursing activities. Format: Face-to-face support sessions. Dose: Six sessions of 1–2 hours. Frequency: Missing. Duration: Three months. Content: Instructor-led situated and simulated commonly occurring clinical situations focusing on recognizing signs and symptoms of diseases, understanding medication regimens, interpret laboratory data, and recognize communication difficulties between the nurses and the patients. Components: Instructor guided participation, questions, and reflection on learning outcomes. | Self-oriented learning. No details about content, format, dose, or duration. | Competence. Post-intervention. |
| 2 | RCT | New nurses. BSc. n: T 239; E 130; C 109 Age (M): 27.5 Gender F: 85% | One hospital, different work units. Sweden Year 2016–2017. | Intervention to increase engagement in proactive behaviors. Format: Group sessions with 10 new nurses and one group leader. Dose: Nine hours plus homework assignments Frequency: Three sessions of 3 hours every other week. Duration: Four weeks. Content: Theory about organizational socialization and proactive behaviors, theory about stress and recovery, group discussions, individual pen-and-paper exercises. Components: Behavior change techniques approach behaviors, systematic exposure, and action planning, homework assignments. | Didactic sessions and skills training. Format: Didactic sessions and skills training. Dose: Nine hours. Frequency: Three sessions (3 hours each) every other week. Duration: Four weeks. Content: Theory and practice on patient care (e.g., nutrition, wound treatment), communication skills, team management, and the role, rights, and responsibilities of nurses. | Role clarity Task mastery Social acceptance Post-intervention. |
| 3 | Observational | New professionals (accountancy, auditing, banking, software, chemical and manufacturing industries) n: T 244 Age (M): 23.6 Gender F: 51% | Unspecified professional settings | On-the-job training (no intervention) | n.a | Role clarity Role conflict Role orientation Four-five months into profession. |
| 4 | Between-groups comparison | New nurses. Different degrees (prelicensure diploma, associate-degree, BSc, or MSc) n: T 763; E 577; C 186 Age (M): 28 Gender F: 91% | Multiple hospitals, different work units. USA Year 2011 | NCSBN's transition-to-practice program. Format: Online course plus preceptorship. Dose: ≥ 20 hours per month for the first 6 months. Frequency: Not specified. Duration: 12 months. Content: Online modules covered the topics patient-centered care, communication and teamwork, evidence-based practice, quality improvement, and informatics. Components: An institution-based orientation program, preceptor for the first six months of practice, educational online modules for the first six months, safety and clinical reasoning threaded throughout the modules, institutional support during the second six months of the program, feedback and reflection. | Original transition-to-practice programs of the health facilities in the control group. No details about content, format, dose, or duration. | Competence (overall) Patient-centred care Evidence-based practice Use of technology Communication and teamwork Post-intervention, 9 and 12-months follow-up. |
| 5 | Between-groups comparison | New nurses. Degree not specified. n: T 332; E 234; C 85 Age (M): 23.5 Gender F: 83.9% | Different hospitals, different work units. Vietnam. Year 2019–2020. | Clinical training program. Format: Clinical training, preceptorship, classroom-based activities Dose: 1520 units. Frequency: Not reported. Duration: Nine months. Content: Clinical training (on-the-job training = ward-based practices involving patients; off-the-job training = classroom-based lectures, discussions, group works, experiments, and practices), preceptor; and systematic management. | Original training program of the health facilities in the control group. No details about content, format, dose, or duration. | Competence (total). Post-intervention. |

Note: Study 1 = Chen et al. (2017); Study 2 = Frögéli et al. (2020); Study 3 = Kowtha (2011); Study 4 = Spector et al. (2015); Study 5 = Horii et al. (2021); n = number of participants; T = total number of participants; E = Experimental group; C = Control group; M = mean; n.a = not applicable.

didactic sessions and skills training [38]. The format, dose, duration, frequency, and content were not reported in three cases [34, 35, 37] and equivalent to the experimental intervention in time, dose, and frequency in one case [38].

Three of the studies included measures of overall competence [34, 35, 37]. In one of these cases a separate measure of specific competencies was also used [34]. One study furthermore included a measure of task mastery [38]. Two studies included a measure of role clarity [36, 38]. Finally, a measure of social acceptance was included in one study [38]. All studies collected baseline data within the first three months following professional entry. In the four intervention studies, the next data collection was at post-intervention (i.e. following the end of the intervention) and in one case [34] follow-up data was also collected at nine and 12 months after baseline. In [36] data for the predictor variable was collected two-three months following professional entry and the dependent variable data was collected at four-five months into the new profession.

### Effects of formal onboarding programs and practices

The objective of this study was to investigate the effectiveness of formal onboarding programs and practices for new professionals. However, the studies investigating effects of onboarding practices were too heterogenous in terms of intervention, comparator, and study design to allow for the conduct of a meta-analysis of effects. The studies investigating onboarding programs were considered homogeneous enough to allow for a meta-analysis based on both clinical, methodological and statistical points of view. That being said, given the fact that there were only two studies of low methodological quality, the analysis would result in a non-generalizable effect estimate of low quality. As the value of such an effect estimate was considered low, we decided not to conduct the meta-analysis. Thus, the results are presented using a narrative summary. The results of the studies are presented in Table 3. For purpose of

**Table 3. Results of included studies and computed effect sizes.**

| Study | Outcome | Pre-intervention | | | | Post-intervention | | | | | | Effect | |
|---|---|---|---|---|---|---|---|---|---|---|---|---|---|
| | | Exp. group | | Con. group | | Time[a] | Exp. group | | Con. group | | | | |
| | | M (sd) | n | M (sd) | n | | M (sd) | n | M (sd) | n | r | d | |
| 1 | Competence | 57.63 (5.14) | 16 | 61.33 (5.08) | 15 | 3 m | 74.81 (6.05) | 16 | 67.93 (3.94) | 15 | | 1.35 | |
| 2 | Task mastery | 3.37 (0.80) | 129 | 3.45 (0.73) | 109 | 7 w | 3.33 (0.91) | 129 | 3.36 (0.73) | 109 | | -0.04 | |
| | Role clarity | 3.71 (0.68) | 129 | 3.75 (0.63) | 109 | 7 w | 3.70 (0.91) | 129 | 3.97 (0.63) | 109 | | -0.41 | |
| | Social acceptance | 3.84 (0.80) | 129 | 3.81 (0.94) | 109 | 7 w | 3.78 (0.91) | 129 | 3.75 (0.94) | 109 | | 0.03 | |
| 3 | Role clarity | | | | | 4–5 m | | | | | 0.40 | 0.87[b] | |
| | Role conflict | | | | | 4–5 m | | | | | -0.41 | -0.90[b] | |
| | Role orientation | | | | | 4–5 m | | | | | -0.02 | -0.04[b] | |
| 4 | Competence (overall) | 2.94 (0.48) | 536 | 3.00 (0.41) | 177 | 6 m | 3.09 (0.38) | 536 | 3.11 (0.41) | 177 | | -0.04 | |
| | Patient-centred care | 3.94 (0.56) | 543 | 3.96 (0.56) | 179 | 6 m | 4.19 (0.52) | 543 | 4.20 (0.50) | 179 | | -0.02 | |
| | Evidence-based practice and quality improvement | 3.72 (0.56) | 542 | 3.70 (0.59) | 179 | 6 m | 4.01 (0.50) | 542 | 3.98 (0.54) | 179 | | 0.05 | |
| | Use of technology | 4.03 (0.59) | 540 | 4.07 (0.57) | 179 | 6 m | 4.32 (0.54) | 540 | 4.31 (0.54) | 179 | | 0.02 | |
| | Communication & teamwork | 3.72 (0.59) | 542 | 3.82 (0.56) | 179 | 6 m | 4.07 (0.51) | 542 | 4.12 (0.48) | 179 | | -0.09 | |
| 5 | Competence (total) | 3.55 (1.22) | 206 | 4.13 (0.93) | 74 | 9 m | 5.16 (0.91) | 206 | 5.01 (0.91) | 74 | | 0.13 | |

Note: Study 1 = Chen et al. (2017); Study 2 = Frögéli et al. (2020); Study 3 = Kowtha (2011); Study 4 = Spector et al. (2015); Study 5 = Horii et al. (2021); Exp. group = experimental group; Con. group = Control group; M = mean; n = number of participants; m = months, w = weeks; r = correlation coefficient; d = Cohens d computed based on pooled sd from baseline

[a]Time from baseline

[b]Computed according to [40]

comparison, the Cohen's d effect size [39] of the between-groups effect at post-intervention in each study has been computed based on pooled standard deviations from baseline. For [36], a measure of Cohen's has been computed based on the correlation between the predictor and dependent variable [40].

In three of the five included studies, a statistically significant effect of onboarding practices and programs could be confirmed [35–37]. Effect sizes ranged from small for the effect of on-the-job training in combination with classroom-based lectures and preceptorship on competence (Cohen's d = 0.13 [35]) to large for the effect of on-the-job training on role clarity (Cohen's d = 0.87 [36]) and the effect of instructor-led situated and simulated training on competence (1.35 [37]). In the remaining two studies that included behavior change techniques for increased engagement in proactive behaviors and reduced stress [38], as well as an online course plus preceptorship [34], no effects of the onboarding practices and programs could be confirmed (Cohen's ds in the range of -0.41 to 0.05). Thus, the results of this study suggest that formal onboarding programs and practices including structured and supported training for new professionals may be effective in improving competence and role clarity. The certainty of the evidence was, however, rated as low indicating that the true effect might be markedly different from the presented [33].

## Discussion

The results of the present review suggest that onboarding strategies and programs with an explicit focus on skills training in day-to-day practice may be effective for improving the adjustment of new professionals. These findings are in line with previous recommendations suggesting that newcomers should be given on-the-job training and be encouraged to monitor how co-workers perform their tasks for skill development [7, 11, 14]. The present review adds a specific focus on new professionals (defined as 18–30 years of age, employed for no longer than three months at start of intervention). In addition, the designs of the studies included in the present review allows for firmer conclusions than previous literature and thus expands the current knowledge of the effect of onboarding practices and programs for new professionals' adjustment.

Although it was not possible to conduct meta-analyses, we were still able to identify studies that evaluated effects of onboarding programs and practices. Large effects of onboarding practices on outcomes of socialization were found in two studies [36, 37]. Interestingly, both onboarding practices investigated focused on skills training. In [37], the intervention included instructor-led training in situated and simulated commonly occurring clinical situations with guided participation, possibility to ask questions, and reflection on learning. In [36] the onboarding practice investigated was on-the-job training which was defined as training that was in-house, specifically designed to give job-related skills, in which each stage of training built upon the experience from the previous stage, and it was clear how one assignment leads to the next assignment. A small effect of an onboarding program on socialization was found in one study [35]. In line with the onboarding practices, there was a focus on on-the-job training here as well. In addition to the training, a preceptorship and classroom-based lectures, discussions, and group assignments were included in the program. In contrast, the onboarding program at trial in [34] that included online didactics and preceptorship but no explicit on-the-job training was not more effective than the comparison condition. Nor was the program in [38] that included group discussions and behavior change techniques focused on stress prevention and increasing proactive behaviors at work (such as asking questions and practicing skills), but not structured and supported on-the-job training.

As the dataset contained four studies on new nurses, it is interesting to compare the results of this review to other reviews where the study population has been restricted to new nurses.

There are over a hundred reviews on different aspects of onboarding/transition to practice programs for new nurses. As far as we know, there are no meta-analyses on the effects of the programs but there are a few reviews presenting effects of programs using narrative synthesis. Considering the research questions of the present review, the most comparable in content are systematic reviews showing effects on learning and performance [41], confidence [42, 43], and social belonging [43]. The authors of all three studies concluded that there is a need for investigations of higher methodological quality, including RCTs and quasi-experimental trials. The present review thus adds to and expands findings of previous reviews within the field of nursing. The authors of [42] concluded that the type of program or strategy is less important, and that it is the attention given to easing the new professionals' transition experiences (as opposed of letting them adjust on their own) that makes a difference. However, the results of the present review may be interpreted to indicate that even though many strategies are probably beneficial, the evidence as of today is stronger for the use of structured and supported on-the-job training than it is for any other strategy.

All of the previous reviews highlight the role of the preceptor as an important support strategy for new nurses [41–43], but also recognize that the role of the preceptor needs to be given more attention and preparation [42]. The results of the present study suggest that one important and empirically validated role of the preceptor could be to support the new professional during structured on-the-job training. This is surely something that most preceptors already do, but perhaps the methods used in such training could be developed with explicit content and a theoretical model of how learning is to come about.

## Methodological quality

The methodological quality of the included studies was rated as low to moderate and there was a high risk of bias. The certainty of the evidence was rated as low indicating that the true effect might be markedly different from the presented. Future research should focus on conducting trials of higher methodological quality that allow for firmer conclusions. This could be RCTs or longitudinal study designs that allows for investigation of within-person effects.

## Limitations

There are a number of shortcomings to the study design that limits the conclusions that may be made. First of all, the review did only consider published (or accepted for publication) papers and no gray literature. Second, the inclusion criteria in relation to the population and intervention were quite narrow, limiting the number of studies that could be included. However, this may also be considered a strength as it makes the limits of the applicability of the findings clear. One group of professionals that were represented in the examined literature but did not make it to the final set of included studies were medical residents. There were many examples of randomized or quasi-experimental studies of interventions (most typically simulation exercises) for new medical professionals. These studies were typically excluded because they were either not interpreted as an onboarding strategy but rather as a very specific skills training strategy (e.g. by targeting residents of different level of training), or the outcome was assessed either using a written test of knowledge or by use of a standardized protocol rated by an observer, none of which were considered appropriate measures of organizational socialization. Others have investigated effects of simulation training for medical residents and found positive meta-analytic post-intervention effects as compared to active and passive interventions on attitudes (including self-efficacy) [44], nontechnical skills (i.e. situation awareness) [45], and performance [46, 47]. As in our investigation, these studies typically included both residents and other populations such as medical students or physicians. However, effects were

no longer present at follow-up [44, 47] and transfer of training was questioned [45–47]. It is suggested that future research should focus on developing theoretically sound interventions focusing on mastery learning and deliberate practice models in simulated and real clinical context. This proposition resonates with the finding in the present review that on-the-job training seems to be the most effective onboarding strategy. Finally, the results of [44–47] suggest that models and interventions of on-the-job training can benefit from studying the training and educational methods used in simulation practice.

The results of the present review are of course also limited to the content of the included interventions. In this sample, on-the-job training stood out as a good strategy for supporting the adjustment of new professionals. However, there are many onboarding strategies in the literature, many of which have great face validity. The results of the presents review should not be interpreted as an indication that these strategies are not effective, but that the evidence as of today is too limited to draw firm conclusions about their effects. Future research will need to broaden this focus and investigate effects of other onboarding strategies in studies with high methodological quality. The same can be said about the optimal format, dose, and frequency of onboarding practices and programs as the available studies are too few to allow for investigations of characteristics. While these limitations are important, by reviewing the literature and summarizing the available data, the present study adds significantly to the current understanding of how to support new professionals' organizational socialization.

## Conclusions and recommendations

In summary, the results of the present review suggest that the structured and supported on-the-job training is the most effective onboarding strategy for supporting new professionals' adjustment, and that it may also successfully be incorporated in onboarding programs. Effects could be confirmed for the adjustment indicators role clarity as well as competence/task mastery. For practitioners, the results suggest that onboarding strategies should prioritize on-the-job training as this is the strategy with the best current evidence for effects. For researchers, the results suggest that attention should be given to understanding how to best implement on-the-job training to ensure strong, broad, and lasting effects. Future research should also investigate effects of on-the-job training on social acceptance as this has not been investigated.

## Supporting information

**S1 Checklist. PRISMA 2020 checklist.**
(DOCX)

**S1 Appendix. Appendix I: Search strategies for included databases.** Appendix II: Studies excluded following full-text investigation with reason for exclusion.
(DOCX)

## Acknowledgments

The authors wish to acknowledge Sabina Gillsund, university librarian, Karolinska Institutet.

## Author Contributions

**Conceptualization:** Elin Frögéli, Petter Gustavsson.

**Data curation:** Elin Frögéli, Bo Jenner, Petter Gustavsson.

**Formal analysis:** Elin Frögéli.

**Funding acquisition:** Petter Gustavsson.

**Investigation:** Elin Frögéli.

**Methodology:** Elin Frögéli, Petter Gustavsson.

**Project administration:** Elin Frögéli.

**Supervision:** Petter Gustavsson.

**Writing – original draft:** Elin Frögéli.

**Writing – review & editing:** Elin Frögéli, Bo Jenner, Petter Gustavsson.

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
