## [Decision Letter · Decision Letter 0]

31 Aug 2022

PONE-D-22-02533Effectiveness of formal onboarding practices and programs versus informal onboarding or ‘treatment as usual’ for facilitating new professionals’ organizational socialization: a systematic reviewPLOS ONE

Dear Dr. Frögéli,

Thank you for submitting your manuscript to PLOS ONE. After careful consideration, we feel that it has merit but does not fully meet PLOS ONE’s publication criteria as it currently stands. Therefore, we invite you to submit a revised version of the manuscript that addresses the points raised during the review process.

We look forward to receiving your revised manuscript.

Kind regards,

Felix Bongomin, MB ChB, MSc, MMed, FECMM

Academic Editor

PLOS ONE

Journal Requirements:

Reviewers' comments:

Reviewer's Responses to Questions

**Comments to the Author**

1. Is the manuscript technically sound, and do the data support the conclusions?

Reviewer #1: Yes

Reviewer #2: Yes

2. Has the statistical analysis been performed appropriately and rigorously? 

Reviewer #1: Yes

Reviewer #2: Yes

3. Have the authors made all data underlying the findings in their manuscript fully available?

Reviewer #1: Yes

Reviewer #2: Yes

4. Is the manuscript presented in an intelligible fashion and written in standard English?

Reviewer #1: Yes

Reviewer #2: Yes

5. Review Comments to the Author

Reviewer #1: Effectiveness of Formal Onboarding Practices and Programs versus informal onboarding or treatment as usual for Facilitating new Professionals’ Organizational Socialization: A systematic Review

REVIEW REPORT

Title

I find the title a bit too long and confusing. A title should be catchy to attract readership. The summary title sounds better than the main title.

• Line 36- What is JBI? Write it in full first then abbreviate subsequently. See also line 37 for GRADE. Read through the document and improve on this.

Abstract

• Abstract is good and straight forward giving a nice summary

• The objectives should be actionable and well formulated.

• All tenses should be in past tense.

Introduction

• Line 65 is it “from” or “form”?

• Line 70-72 though complete but needs to be followed by another comment as to whether the goal is being fulfilled or not. As it stands, it is hanging.

• Line 80-83 is incomplete.

• Line 83 is also incomplete; the meta-analysis was by?

• In line 89, the authors have written in full the IWG model and followed by explaining what it is but failed to explain what the 4 Cs model stands for. Though not broadly discussed but it is important to explain what the 4 Cs are, do not assume that every reader knows.

• Generally, the introduction is clear and specific to the subject under study.

Objectives

• Should the definition of terms be in the objectives section?

• The authors have the purpose in line 153 but not the objective, or are they one and the same thing? Remember the objectives give vision to your study so they should stand out and be clearly stated as much as possible.

• What is PROSPERO? What about JBI? Write them in full first then abbreviate going forward.

• Some of the definitions of terms are not clear eg ‘Treatment as usual’. Authors should give the definition as used in the study and how it was measure/ applied.

Inclusion Criteria

• The authors have talked about those who were excluded, so who were included given that in this section you are talking about the inclusion criteria. There is a mix-up on the exclusion criteria, so either the authors change the sub-title to read ‘exclusion and inclusion criteria’ or discuss more the inclusion criteria as guided by the subtitle.

Methods

This section is detailed and well discussed. Search selection, study screening and selection as well as the critical appraisal are well presented, clear and straight forward.

Data extraction

• The section has a lot of repetition revise to minimize on repetition and make the section clearer.

• Exactly how was data extracted? This is not clear. The authors need to give a blue print of the process, this has not come out clearly in the explanation given.

Data synthesis and assessment of certainty in the findings

• It is important to explain further how GRADE ratings were used. This is because the purpose of the methodology section is to give a step-by-step process to make it as clear as possible so that somebody else can use it to carry out a similar study elsewhere.

Results

• Line 287 & 288 -Why should authors follow a subtitle with another subtitle without saying anything under the results subsection to necessitate another subtitle.

• Study inclusion- This subtitle is a bit confusing and the reader might think that the authors are talking about the inclusion criteria as above when it is not the case.

• The authors might want to change to a relevant sub-title or simply leave the section directly under the results sub heading.

Methodological Quality

• Stick to past tense.

• This section is clear with the process being clearly discussed in detail. There is however very little on findings. More details on the findings should be presented and the tables interpreted. There is also need to state the implications of the findings and relate it to the objectives of the study. This will bring clarity to the findings.

Discussion

• This section is well done and it discusses clearly both practices and programs in line with the purpose of the study.

• There is need in the discussion however to mention the importance of this approach to knowledge mining as compared to mining of primary data.

Conclusion

A good and appropriate conclusion has been given

• Line 531- is it founding or funding?

References

References are well done but need but with very many typographical errors. The authors should be keen on referencing protocols

Reviewer #2: PONE-D-22-02533: Effectiveness of formal onboarding practices and programs versus informal onboarding or ‘treatment as usual’ for facilitating new professionals’ organizational socialization: a systematic review

Overview

The manuscript is well written and organized with good flow. The rigorous screening of publications using predefined criteria is commendable. This process, however, resulted in the exclusion of thousands of empirical studies that could have enriched the discussion and introduce further cross-cultural perspectives.

Abstract

In the abstract, I would recommend to include the ‘inclusion criteria’ within the ‘methods’ section rather than standing separately.

Introduction

I forward the same recommendation regarding ‘inclusion criteria’ in the main body of the manuscript. In addition, it would make more sense if the objective section does not stand alone but comes as a wrapping up of the introduction section that drew heavily on the critical review of the relevant literature.

The results

The results section could have been enriched with presentation of findings on the proposed links between onboarding practices and programs, on the one hand, and role clarity, task mastery, and social acceptance, on the other hand. The studies must have included explanatory hypotheses on how the associations between the independent and outcome variables come about.

The discussion

I appreciate the honesty and modest conclusions and insights the authors make based on the results of the systematic review, esp. with four of the five studies covered in the review covering nurses as study populations.

Conclusions and recommendations

As I mentioned at the beginning, the manuscript would have been enriched with discussion on contextual and explanatory propositions with broader applicability had it included observational studies. But that is one of the drawbacks of being stringent with the inclusion criteria. Besides, the figure that describes the screening process, the reasons for the exclusion of some of the studies were not clearly stated. It would help if that information was provided as the authors did for the rest of the methodological issues.

6. PLOS authors have the option to publish the peer review history of their article (what does this mean?). If published, this will include your full peer review and any attached files.

Reviewer #1: No

Reviewer #2: **Yes: **Mikyas Abera

---

## [Author Response · Author response to Decision Letter 0]

22 Sep 2022

A separate file is attached with responses to the reviewers.

---

## [Decision Letter · Decision Letter 1]

1 Feb 2023

Effectiveness of formal onboarding for facilitating organizational socialization: a systematic review

PONE-D-22-02533R1

Dear Elin Frögéli,

We’re pleased to inform you that your manuscript has been judged scientifically suitable for publication and will be formally accepted for publication once it meets all outstanding technical requirements.

Kind regards,

Tai Ming Wut

Academic Editor

PLOS ONE

Additional Editor Comments (optional):

Reviewers' comments:

Reviewer's Responses to Questions

**Comments to the Author**

1. If the authors have adequately addressed your comments raised in a previous round of review and you feel that this manuscript is now acceptable for publication, you may indicate that here to bypass the “Comments to the Author” section, enter your conflict of interest statement in the “Confidential to Editor” section, and submit your "Accept" recommendation.

Reviewer #1: All comments have been addressed

Reviewer #2: All comments have been addressed

2. Is the manuscript technically sound, and do the data support the conclusions?

Reviewer #1: Yes

Reviewer #2: Yes

3. Has the statistical analysis been performed appropriately and rigorously? 

Reviewer #1: Yes

Reviewer #2: Yes

4. Have the authors made all data underlying the findings in their manuscript fully available?

Reviewer #1: Yes

Reviewer #2: Yes

5. Is the manuscript presented in an intelligible fashion and written in standard English?

Reviewer #1: Yes

Reviewer #2: Yes

6. Review Comments to the Author

Reviewer #1: After addressing all the concerns i had previously, the article now makes a lot of sense and ready for publication as it is.

Reviewer #2: The authors have addressed all the comments and suggestions that I made on their draft manuscript. Save for the concerns of the other reviewer and editorial issues, I would recommend the manuscript for publication.

7. PLOS authors have the option to publish the peer review history of their article (what does this mean?). If published, this will include your full peer review and any attached files.

Reviewer #1: No

Reviewer #2: **Yes: **Mikyas Abera

---

## [Editor Report · Acceptance letter]

7 Feb 2023

PONE-D-22-02533R1 

Effectiveness of formal onboarding for facilitating organizational socialization: a systematic review 

Dear Dr. Frögéli:

I'm pleased to inform you that your manuscript has been deemed suitable for publication in PLOS ONE. Congratulations! Your manuscript is now with our production department. 

Kind regards, 

on behalf of

Dr. Tai Ming Wut 

Academic Editor

PLOS ONE